

# Melatonin increases growth properties in human dermal papilla spheroids by activating AKT/GSK3β/β-Catenin signaling pathway

Sowon Bae[1], Yoo Gyeong Yoon[1,2], Ji Yea Kim[3], In-Chul Park[3], Sungkwan An[1], Jae Ho Lee[1] and Seunghee Bae[1]

[1] Research Institute for Molecular-Targeted Drugs, Department of Cosmetics Engineering, Konkuk University, Seoul, Republic of Korea
[2] R&D Planning Dept., Dermalab Co., Ltd, Suwon-si, Gyeonggi-do, Republic of Korea
[3] Division of Fusion Radiology Research, Korea Institute of Radiological & Medical Sciences, Seoul, Republic of Korea

Corresponding author
Seunghee Bae, sbae@konkuk.ac.kr

## ABSTRACT

**Background**. Melatonin, a neurohormone, maybe involved in physiological processes, such as antioxidation, anti-inflammation, and hair growth. In the present study, we investigated the effects of melatonin on proliferation and intracellular signaling in DP cells using a three-dimensional (3D) spheroid culture system that mimics the *in vivo* hair follicle system.

**Methods**. DP cells were incubated in monolayer (2D) and 3D spheroid culture systems. The expression levels of melatonin receptors in DP cells were analyzed using quantitative reverse transcription polymerase chain reaction (qRT-PCR) and western blotting. The effect of melatonin on the hair-inductive property of DP cells was analyzed using a WST-1-based proliferation assay, determination of DP spheroid size, expression analysis of DP signature genes, and determination of β-catenin stabilization in DP cells. The AKT/GSK3β/β-catenin signaling pathway associated with melatonin-induced β-catenin stabilization in DP cells was investigated by analyzing changes in upstream regulator proteins, including AKT, GSK3β, and their phosphorylated forms.

**Results**. The expression levels of the melatonin receptors were higher in human DP cells than in human epidermal keratinocytes and human dermal fibroblast cells. Comparing the expression level according to the human DP cell culture condition, melatonin receptor expression was upregulated in the 3D culture system compared to the traditional two-dimensional monolayer culture system. Cell viability analysis showed that melatonin concentrations up to 1 mM did not affect cell viability. Moreover, melatonin increased the diameter of DP cell 3D spheroids in a dose-dependent manner. Immunoblotting and qRT-PCR analysis revealed that melatonin upregulated the expression of hair growth-related genes, including alkaline phosphatase, bone morphogenetic protein 2, versican, and wingless-int 5A, in a melatonin receptor-dependent manner. Cell fractionation analysis showed that melatonin increased the nuclear localization of β-catenin. This result correlated with the increased transcriptional activation of T-cell factor/lymphoid enhancer factor-responsive luciferase induced by melatonin treatment. Interestingly, melatonin induced the phosphorylation of protein kinase B/AKT at serine 473 residue and GSK-3β at serine 9 residue. To determine whether AKT phosphorylation at serine 473 induced β-catenin nuclear

translocation through GSK3β phosphorylation at serine 9, the PI3K/AKT inhibitor LY294002 was cotreated with melatonin. Immunoblotting showed that LY294002 inhibited melatonin-induced phosphorylation of GSK3β at serine 9 residue and β-catenin activation.

**Conclusion**. Collectively, this report suggests that melatonin promotes growth properties by activating the AKT/GSK3β/β-catenin signaling pathway through melatonin receptors.

# INTRODUCTION

Hair serves as biological protection from the external environment and influences social interactions (*Cruz et al., 2016*). The growth and shape of hair fibers are tightly controlled and generated by hair follicles (HFs), which are specialized miniature organs that anchor each hair to the skin (*Westgate, Botchkareva & Tobin, 2013*). HFs are composed of the root sheath (HF coat region), bulge (terminal part of the HF), papilla (base region of the hair bulb), and matrix (around the papilla region) (*Bukvić Mokoš & Mosler, 2014*). HFs undergo characteristic phases known as the hair cycle, which includes the anagen (growth), catagen (regression), telogen (resting), and regeneration stages (*Alonso & Fuchs, 2006*). In particular, dermal papilla (DP) cells originate from blimp1+ fibroblasts (dermal stem cells) during embryonic development (*Millar, 2002*), provide instructive signals required to activate epithelial progenitors involved in hair formation, proliferation, and differentiation, and initiate HF regeneration (*Enshell-Seijffers et al., 2010*; *Paus & Cotsarelis, 1999*; *Sennett & Rendl, 2012*). Moreover, DP cell numbers fluctuate with the hair cycle, and hair loss is closely related to gradual depletion and atrophy of DP cells (*Rahmani et al., 2014*). In patients with alopecia, hair follicles are miniaturized, and the rate of hair growth induction by DP is reduced (*Santos, Avci & Hamblin, 2015*). Therefore, several studies have focused on elucidating the mechanism of hair loss and identifying therapeutic candidates using human DP cells (*Madaan et al., 2018*).

Winless-(WNT)/β-catenin signaling plays a critical role in HF morphogenesis and regeneration (*Andl et al., 2002*; *Enshell-Seijffers et al., 2010*; *Lowry et al., 2005*). In DP cells, ablation of the β-catenin gene results in premature induction of the catagen phase and prevents normal regeneration of HFs from stem cells (*Enshell-Seijffers et al., 2010*). Therefore, WNT/β-catenin signaling is critically involved in anagen phase maintenance and hair induction in DP cells (*Kishimoto, Burgeson & Morgan, 2000*; *Soma et al., 2012*). When this pathway is activated, stabilized β-catenin is translocated to the nucleus and interacts with T-cell factor/lymphoid enhancer factor (TCF/LEF) transcription factors to transcribe target genes, such as LEF1 (*DasGupta & Fuchs, 1999*). However, DP cells lose their ability to induce HF neogenesis during passage in a two-dimensional (2D) monolayer culture system (*Ohyama et al., 2010*; *Yang & Cotsarelis, 2010*), whereas three-dimensional

(3D) spheroid DP cells restore the transcriptional activity of signature genes and their ability to induce de novo HFs (*Higgins et al., 2013*; *Kang et al., 2012*; *Shimizu et al., 2011*). Therefore, 3D-culture systems have been frequently used as *in vitro* models to investigate hair growth (*Choi et al., 2017*; *Lee et al., 2019*).

Melatonin (N-acetyl-methoxy-tryptamine), an indole-like neurohormone, is mainly produced in the human pineal gland with a circadian rhythm (*Reiter, 1991*). The cyclic synthesis of melatonin regulates various physiological responses, such as seasonal biorhythms and daily sleep-wake cycles (*Cardinali & Pévet, 1998*). Further studies have demonstrated that melatonin has strong antioxidant properties that can help protect against cellular damage caused by oxidative stress and aging processes (*Reiter et al., 2003*; *Tamura et al., 2012*; *Tan et al., 2002*). Additionally, several recent studies have reported that melatonin can be synthesized and metabolized in extra-pineal regions, especially in the skin (*Slominski et al., 2008*). The biological functions of melatonin are mediated by its receptors, melatonin receptor 1 (MTNR1A) and melatonin receptor 2 (MTNR1B) (*Dubocovich et al., 2003*). Interestingly, melatonin receptors are expressed in keratinocytes, melanocytes, and dermal fibroblasts (*Slominski et al., 2003*; *Slominski, Wortsman & Tobin, 2005*). In skin cells, these receptors are involved in cellular proliferation and differentiation (*Slominski, Wortsman & Tobin, 2005*). Moreover, it was reported that HFs are a target site for melatonin synthesis (*Fischer et al., 2008*; *Kobayashi et al., 2005*). Studies have shown that the expression of melatonin receptor proteins in the HFs is dependent on the hair cycle, and melatonin concentration in human scalp HFs is reportedly higher than the levels in the serum, suggesting that melatonin may potentially regulate hair growth (*Kobayashi et al., 2005*). Several studies have demonstrated the effects of melatonin on the regulation of hair growth. In a murine model, treatment with melatonin inhibited HF keratinocyte apoptosis and downregulated the expression of estrogen receptors expressed in late telogen at maximal levels (*Uslu et al., 2014*). Furthermore, in an *in vitro* human HF culture model, treatment with melatonin improved the rate of hair follicle growth, and this effect was suppressed by a potent melatonin antagonist (*Fischer et al., 2000*). Additionally, a double-blind, placebo-controlled pilot study revealed that topical treatment with 0.1% melatonin in patients with alopecia resulted in a significant increase in anagen hairs compared with placebo (*Fischer et al., 2004*). It was also reported that the effect of melatonin on human hair growth improved when melatonin was formulated into a nanostructured lipid carrier epidermal delivery system (*Hatem et al., 2018*). These results suggest that melatonin exerts potent effects on hair growth and is a potential pharmacological candidate for preventing hair loss symptoms. However, the effects of melatonin and its receptors on the proliferation of DP cells and the underlying mechanisms remain unknown. In the present study, we investigated the effects of melatonin on proliferation and intracellular signaling in human DP cells using a 3D spheroid culture system to mimic *in vivo* HF conditions.

## MATERIALS & METHODS

### Cell culture and reagents

Three to seven human dermal papilla (HDP) passage cells (PromoCell, Heidelberg, Germany) were grown in 5% $CO_2$ at 37 °C using a follicle DP cell growth medium kit

(PromoCell). For experiments, cells were seeded in Dulbecco's modified Eagle's medium (DMEM; Thermo Fisher Scientific, Waltham, MA, USA) supplemented with 5% (v/v) fetal bovine serum (FBS) and 1% penicillin/streptomycin (Thermo Fisher Scientific). Human epidermal keratinocyte (HEK) cells (CELLnTEC, Bern, Switzerland) were maintained in CnT-Prime Epithelial Culture Medium (CELLnTEC) in 5% $CO_2$ at 37 °C. Human dermal fibroblast (HDF) cells (Lonza, Basel, Switzerland) were maintained in DMEM (Thermo Fisher Scientific) supplemented with 10% (v/v) FBS and 1% penicillin/streptomycin (Thermo Fisher Scientific). 293T cells (American Type Culture Collection, Manassas, VA, USA) were maintained in DMEM supplemented with 10% (v/v) FBS (Thermo Fisher Scientific) and 1% penicillin/streptomycin (Thermo Fisher Scientific). Antibodies against lamin A/C (sc-7293) and β-tubulin (sc-5274) were purchased from Santa Cruz Biotechnology (Dallas, TX, USA). MTNR1A (ab203038), MTNR1B (ab203346), bone morphogenetic protein 2 (BMP2; ab14933), versican (VCAN) (ab19345), and alkaline phosphatase (ALP) (ab108337) were purchased from Abcam (Cambridge, UK). Antibodies against protein kinase B (AKT) (#9272), phosphorylated AKT at serine 473 residue (#9271), phosphorylated AKT at threonine 308 residue (#9275), protein kinase A (PKA) (#4782), phosphorylated PKA at threonine 198 residue (#4781), extracellular signal-regulated kinase (ERK) (#9102), phosphorylated ERK at threonine 202/tyrosine 204 residues (#9101), p38 (#9212), phosphorylated p38 at threonine 180/tyrosine 182 residues (#9211), SRC (#2108), phosphorylated SRC at tyrosine 416 residue (#2101), glycogen synthase kinase 3β (GSK-3β) (#9315), phosphorylated GSK-3β at serine 9 residue (#9323), β-catenin (#9562S), phosphorylated β-catenin at serine 33/37/threonine 41 residues (#9561S), and WNT5a (#2392) were purchased from Cell Signaling Technology (Beverly, MA, USA). Antibodies against β-actin were purchased from Sigma-Aldrich (Darmstadt, Germany USA). TCF/LEF luciferase reporter plasmid was purchased from Promega (Madison, WI, USA). The selective phosphatidylinositol 3-kinase (PI3K) inhibitor LY294002, melatonin, and the melatonin receptor antagonist luzindole were purchased from Sigma-Aldrich. In the present study, melatonin was prepared as a 500 mM stock solution with dimethyl sulfoxide (DMSO) solvent and was used at DMSO concentrations up to 0.2%.

## Cell viability analysis

Cell viability analysis was performed using EZ-Cytox (ITSbio, Seoul, South Korea) and a water-soluble tetrazolium salt (WST-1) assay kit, according to the manufacturer's protocol. Eighty thousand HDP cells were seeded in clear 96-well flat-bottom ultra-low attachment microplates (Corning, Corning, NY, USA) and further cultured for 24 h. The cells were treated with the indicated concentrations of melatonin (0, 0.1, 0.25, 0.5, 0.75, 1, 2.5, and 5 mM) and incubated for 48 h. The WST-1 solution was diluted to 10% in medium and subsequently added to each well. Cell viability was assessed by measuring absorbance at 450 nm using a Synergy$^{TM}$ HTX Multi-Mode Microplate Reader (Bioteck, Winooski, VT, USA).

### 3D spheroid culture of HDP cells

3D spheroid culture of HDP cells was performed as previously described (*Choi et al., 2017*; *Lee et al., 2019*). Fifty thousand HDP cells were seeded in clear 96-well round-bottom ultra-low attachment microplates (Corning, Glendale, AZ, USA) and maintained for 24 h to obtain one spherical cell structure. After incubation, spheroid cells were treated with the indicated concentrations of melatonin (0, 0.5, 0.75, and 1 mM) and further incubated for 48 and 96 h. The diameters of the spheroids were measured using phase-contrast microscopy.

### Luciferase reporter assay

293T cells ($1 \times 10^5$ cells/well) were seeded and cultured for 24 h. The cells were then co-transfected with the TCF/LEF promoter-driven luciferase reporter plasmid and pSV- β-galactosidase (pSV-β-gal) plasmid using Lipofectamine reagent (Thermo Fisher Scientific). Four hours post-transfection, cells were treated with melatonin (0, 0.75, and 1 mM) alone or with LY294002 (Sigma-Aldrich). At 0, 12, and 24 h post-treatment, transfected cells were lysed using passive lysis buffer (Promega). To determine luminescence, the cell lysates were incubated with d-luciferin (Sigma-Aldrich) and luciferase activity was measured using a Synergy HTX Multi-Mode Microplate Reader (Bioteck). Beta-galactosidase activity was measured using the Luminescent β-galactosidase detection kit II (Takara Bio Inc., Tokyo, Japan). Relative luciferase activity was calculated by normalizing luciferase activity to β-galactosidase activity.

### Quantitative reverse transcription-polymerase chain reaction (qRT-PCR) analysis

Total RNA was extracted using TRIzol reagent (Thermo Fisher Scientific), and cDNA was synthesized from 2 μg of the total RNA using oligo dT primers, 1.5 mM dNTPs, 0.1 M DTT, 5X First-Strand Buffer, and M-MLV reverse transcriptase (Thermo Fisher Scientific). qRT-PCR was carried out using a StepOnePlus Real-Time PCR system (Thermo Fisher Scientific) and detected using SYBR Green PCR Master Mix (Thermo Fisher Scientific). The expression levels of mRNA were quantified using the $2^{-\Delta\Delta Ct}$ method and normalized to the expression level of the glyceraldehyde-3-phosphate dehydrogenase (*GAPDH*) housekeeping gene, an internally expressed reference gene. The following primers were used for the amplification of specific genes: WNT family member 5A (*WNT5A*), 5′-TTGAAGCCAATTCTTGGTGGTCGC-3′(forward) and 5′-TGGTCCTGATACAAGTGGCACAGT-3′(reverse); *ALP*, 5′-CAAACCGAGATACAAGCACTCCC-3′(forward) and 5′-CGAAGAGACCCAATAGGTAGTCCAC-3′(reverse); *VCAN*, 5′-GGCAATCTATTTACCAGGACCTGAT-3′(forward) and 5′-TGGCACACAGGTGCATACGT-3′(reverse); *BMP2*, 5′-GGAACGGACATTCGGTCCTT-3′(forward) and 5′-CACCATGGTCGACCTTTAGGA-3′(reverse); *MTNR1A*, 5′-GCCACAGTCTCAAGTACGACA-3′(forward) and 5′- CTGGAGAACCAGGATCCATAAT-3′(reverse); *MTNR1B*, 5′- TACGACCCACGCATCTATTCC- 3′(forward) and 5′-AGGTAGCAGAAGGACACGACA- 3′(reverse); and *GAPDH*, 5′-CGGAGTCAACGGATTTGGTCGTAT-3′(forward) and 5′-AGCCTTCTCCATGGTGAAGAC-3′(reverse).

## Cell fractionation analysis

Total cell lysates were prepared and protein extracts were obtained after lysis in RIPA buffer (25 mM Tris-Cl, 5 mM ethylenediaminetetraacetic acid , 150 mM NaCl, 1% NP40, 1% sodium deoxycholate, 0.025% sodium dodecyl sulfate) containing 200 mM phenylmethylsulfonyl fluoride. Nuclear and cytoplasmic extraction reagents (Thermo Fisher Scientific) were used for nuclear and cytoplasmic fractionation of total cell lysates according to the manufacturer's protocol.

## Statistical analysis

All statistical analyses were performed in triplicate. All data are presented as the mean $\pm$ standard deviation. Normally distributed data were analyzed using GraphPad Prism (v8.4) (GraphPad Software, San Diego, CA, USA). For the results, the $p$-value is from a one-way analysis of variance (ANOVA) followed by Turkey's post hoc test or two-way ANOVA followed by Turkey's post hoc test. Cell viability data were analyzed using a two-tailed Student's $t$-test. Statistical significance was set at $P < 0.05$.

# RESULTS

## Differential expressions of melatonin receptors in human DP cells cultured in conventional 2D monolayer culture and 3D spheroid culture

We examined the expression of melatonin receptors in primary skin cells to study melatonin-induced intracellular signaling in skin cells, including HDP cells. We analyzed the expression levels of MTNR1A and MTNR1B in HEK, HDF, and HDP cells. As shown in Fig. 1A, the transcriptional expression levels of MTNR1A and MTNR1B in HDP cells were higher than that in HEK and HDF cells. Similarly, immunoblotting showed that protein expression was higher in HDP cells (Fig. 1B). We further compared the expression of melatonin receptors in HDP cells under conventional 2D monolayer culture and 3D sphere culture conditions, which mimics the actual *in vivo* environment. As shown in Fig. 1C, the transcriptional expression of melatonin receptors, especially MTNR1B, was higher in the 3D sphere culture condition than in the 2D monolayer culture condition. These results show that 3D sphere culture conditions upregulated the expression of melatonin receptor in HDP cells.

## Melatonin increases the volume of 3D spheroid of human DP cells

As the expression of melatonin receptors was examined in 3D sphere-cultured HDP cells, we analyzed the effect of melatonin on the toxicity and proliferation of 3D sphere-cultured HDP cells using the WST-1 assay. Sphere-cultured HDP cells were treated with 0.1, 0.25, 0.5, 0.75, 1, 2.5, and 5 mM melatonin for 48 h, and cell viability was determined using the WST-1 assay. As shown in Fig. 2A, cell viability was not significantly affected by melatonin at concentrations up to 1 mM in sphere-cultured HDP cells. However, treatment with 2.5 and 5 mM melatonin significantly reduced HDP cell viability to 12.7% and 21.9%, respectively, suggesting that melatonin at a concentration of 1 mM or higher affected the viability of HDP cells (Fig. 2A). We then measured and compared the diameters of spheroids at 48 h

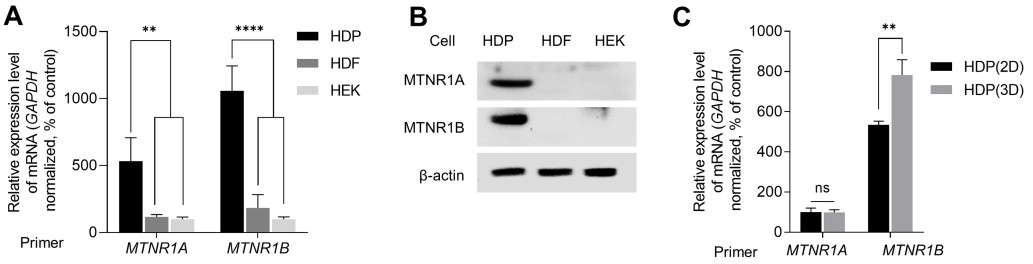

**Figure 1 Expression of melatonin receptors in human dermal papilla (DP) cells.** Human epidermal keratinocyte (HEK), human dermal fibroblast (HDF), and human dermal papilla (HDP) cells were incubated for 48 h. (A) The mRNA levels of *MTNR1A* and *MTNR1B* were assessed by quantitative reverse transcription polymerase chain reaction (qRT-PCR), and *GAPDH* served as an endogenous control. (B) Protein levels of MTNR1A and MTNR1B were assessed by western blotting, and β-actin served as a loading control. (C) HDP cells were incubated for 48 h in 2D and 3D culture systems, and mRNA levels of *MTNR1A* and *MTNR1B* were assessed by qRT-PCR. *GAPDH* served as an endogenous control. Data are presented as the mean ± standard deviation (SD) of three independent experiments, and normally distributed data were evaluated using two-way analysis of variance (ANOVA), followed by Tukey's post hoc test; ** $p < 0.01$, **** $p < 0.0001$ *versus* HDP control cells ($p = 0.0019$, $p = 0.0014$, $p < 0.0001$, and $p < 0.0001$, respectively) (A), and ** $p < 0.01$ *versus* 2D cultured HDP control cells ($p = 0.0052$) (C).

and 96 h to analyze the effect of melatonin on spheroid proliferation (Fig. 2B). It has been reported that sphere formation of DP cells is closely related to their hair-inductive properties; DP sphere size is directly linked to hair shaft diameter (*Morgan, 2014*; *Shimizu et al., 2011*). Figures 2B and 2C show that after 96 h, the size of sphere-cultured HDP cells increased by 12.25%, 20.29%, and 20.86% after treatment with 0.5, 0.75, and 1 mM melatonin, respectively. These data suggest that melatonin has proliferation and formation abilities in the HDP spheroids.

## Melatonin induced hair growth properties are mediated by melatonin receptor pathway

Several studies have indicated that DP signature genes, including ALP, VCAN, BMP2, and wingless-type MMTV integration site family member 5A (WNT5A), are important regulators of hair inductivity *in vivo* (*Iida, Ihara & Matsuzaki, 2007*; *McElwee et al., 2003*; *Ohyama et al., 2010*; *Reddy et al., 2001*; *Rendl, Polak & Fuchs, 2008*; *Soma, Tajima & Kishimoto, 2005*; *Yang & Cotsarelis, 2010*). Therefore, we examined whether melatonin affects the expression of DP signature genes. Sphere-cultured HDP cells were treated with 0.75, and 1 mM melatonin for 48 h, and the expression levels of DP signature genes were determined using qRT-PCR. As shown in Figs. 3A, 3B, 3C, and 3D, melatonin increased the mRNA expression of DP signature genes *ALP*, *VCAN*, *BMP2*, and *WNT5A* in a dose-dependent manner compared with the untreated control. Moreover, immunoblotting showed that the protein expression levels of these genes were increased with melatonin treatment (Fig. 3E). To determine whether the increase in the expression of DP signature genes by melatonin is due to melatonin receptor signaling, we investigated the effect of melatonin-induced expression of DP signature genes following treatment with the melatonin receptor antagonist luzindole. Sphere-cultured HDP were cotreated with 1 mM melatonin alone or 50 μM luzindole for 48 h and the expression of *BMP2* and *WNT5A* was

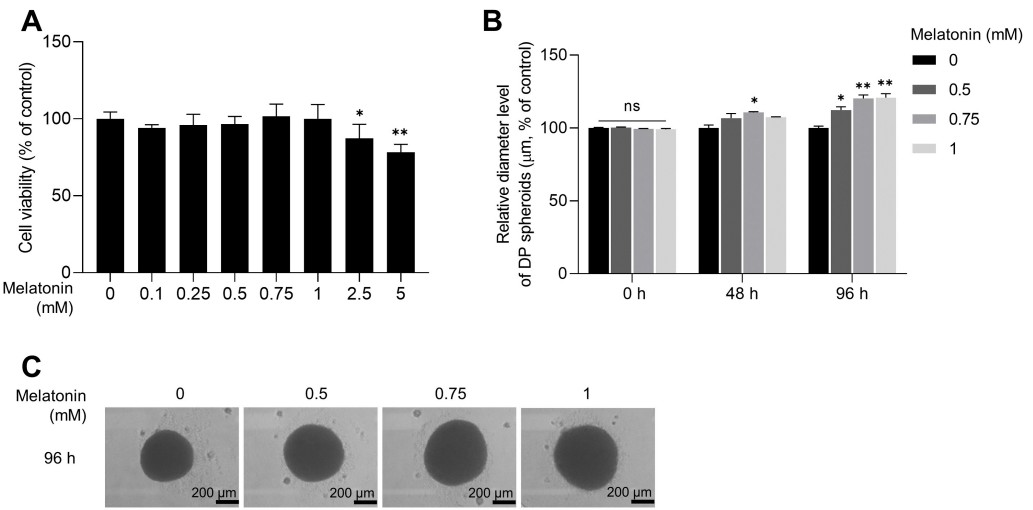

**Figure 2 Effect of melatonin on cell viability and spheroid size of HDP cells.** (A) HDP cells were treated with melatonin (0–5 mM) for 48 h. Cell cytotoxicity was determined using a WST-1 assay. (B) A 96-well clear round-bottom ultra-low attachment microplate was used to measure the size of HDP spheroids. Cells were treated with 0.5, 0.75, and 1 mM melatonin for 48 h and 96 h. Data are presented as the mean ± SD of three independent experiments. (C) Phase-contrast images of spheroids were captured after 96 h. (A) Two-tailed Student's *t*-test; *$p < 0.05$, ** $p < 0.001$ *versus* DMSO treated control ($p = 0.0228$, and $p = 0.000106$ respectively). (B) Two-way ANOVA, followed by Turkey's post hoc test: *$p < 0.05$, **$p < 0.01$ *versus* DMSO treated control (48 h, $p = 0.0237$; 96 h, $p = 0.0101$, $p = 0.0025$, and $p = 0.0016$ respectively). (C) Scale bars represent 200 μm.

examined by qRT-PCR and immunoblotting. The expression of *WNT5A* and *BMP2*, which increased following melatonin treatment, decreased at the transcriptional (Figs. 3F and 3G) and protein levels (Fig. 3H) following cotreatment with luzindole. Collectively, these results suggest that melatonin has the potential to effectively upregulate hair-inductive properties by promoting the expression of DP signature genes that regulate melatonin receptor signaling in HDP spheroids.

## Melatonin activates the WNT/β-catenin signaling pathway in 3D HDP spheroids

WNT/β-catenin signaling is a master regulator of hair-inductive properties of DP cells (*Enshell-Seijffers et al., 2010*; *Kishimoto, Burgeson & Morgan, 2000*; *Lowry et al., 2005*; *Soma et al., 2012*). Therefore, we investigated whether melatonin treatment activated WNT/β-catenin signaling in HDP spheroids. As shown in Fig. 4A, melatonin decreased the phosphorylation of β-catenin at serine 33/37 residues and increased the total protein level of β-catenin in a dose-dependent manner. Moreover, increase in β-catenin protein levels and TCF/LEF reporter activity was also observed when DP spheroids were treated with 10 nM and 10 μM melatonin for a longer duration (72 h), suggesting that the effects of melatonin on the growth of DP spheroids is not the result of high treatment concentration (1 mM) (Fig. S1). Cell fractionation showed that β-catenin translocation into the nucleus increased with melatonin treatment in HDP spheroids compared to that in DMSO-treated controls (Fig. 4B). Next, we analyzed whether TCF/LEF transcriptional activity was upregulated

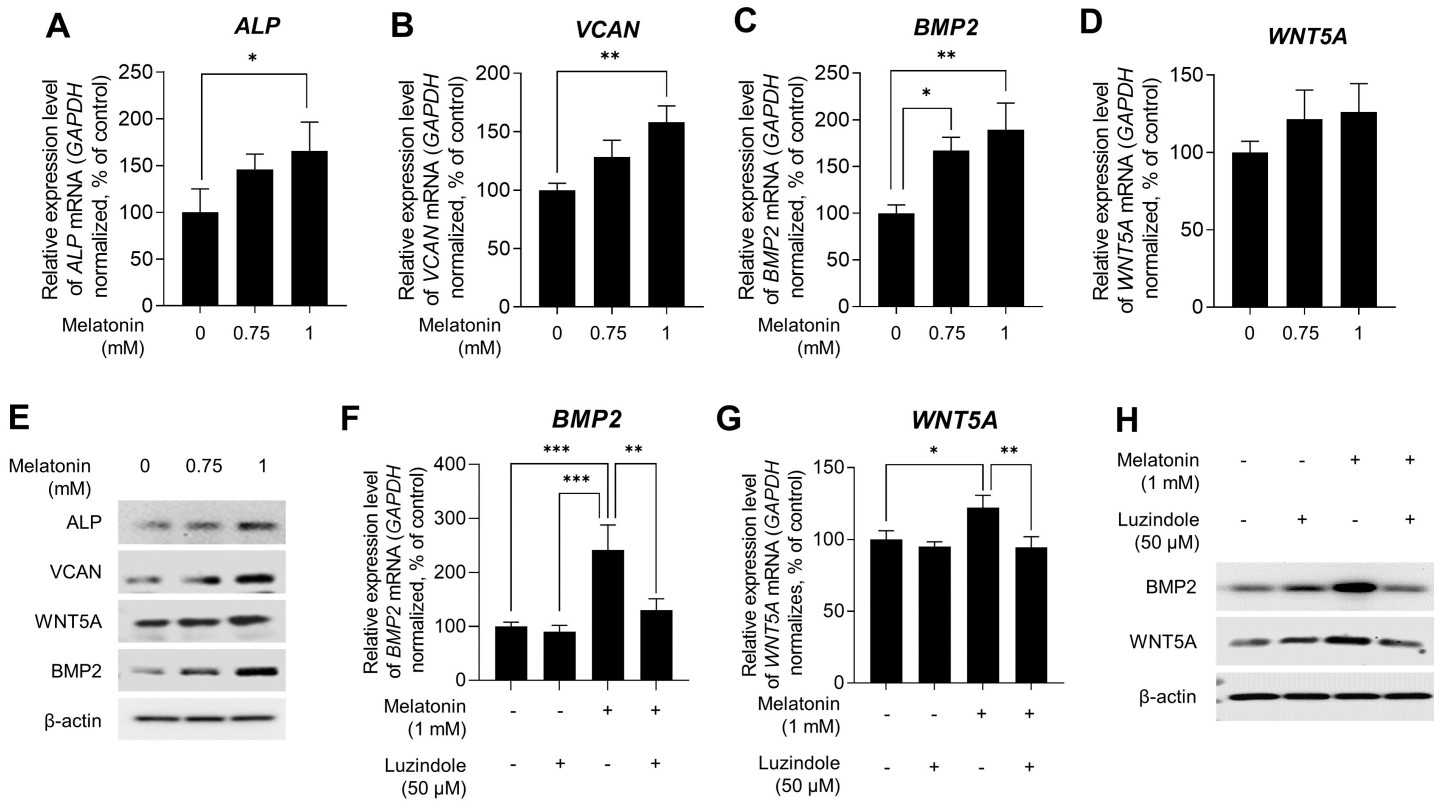

**Figure 3** **Effect of melatonin receptors pathway on hair growth properties genes & protein in HDP spheroids.** (A–E) HDP cells were treated with 0.75 and 1 mM melatonin for 48 h. (A–D) mRNA levels of *ALP, VCAN, BMP2,* and *WNT5A* were assessed by qRT-PCR and normalized against *GAPDH*. (E) Protein levels of these signature genes were assessed by western blotting and β-actin served as a loading control. (F–H) HDP cells were cotreated with 1 mM melatonin with or without 50 μM luzindole for 48 h. (F and G) mRNA levels of *WNT5A* and *BMP2* were assessed by qRT-PCR and *GAPDH* served as an endogenous control. (H) Protein levels of WNT5A and BMP2 were assessed by western blotting and β-actin served as a loading control. Data are presented as the mean ± SD of three independent experiments. (A–D) One-way ANOVA, followed by Turkey's post hoc test; * $p < 0.05$, ** $p < 0.01$, *** $p < 0.001$ *versus* DMSO treated control (A, $p = 0.0401$; B, $p = 0.0025$; C, $p = 0.012$, and $p = 0.003$ respectively), and *versus* melatonin alone treatment control (F, $p = 0.0008$, $p = 0.0005$, and $p = 0.0038$; G, $p = 0.0428$, and $p = 0.0058$ respectively).

by melatonin treatment in 293T cells expressing melatonin receptors (*Chan et al., 1997*). Luminescence analysis showed that melatonin upregulated TCF/LEF-driven luciferase activity in a dose-and time-dependent manner (Fig. 4C).

## AKT signaling mediated the activation of melatonin-induced GSKSβ/β-catenin signaling pathway in 3D HDP spheroids

GSK3β is a major regulator of the canonical WNT signaling pathway (*Wu & Pan, 2010*). Therefore, we investigated the effect of melatonin on kinase that regulate GSK3β. After HDP spheroids were treated with 1 mM melatonin for 24 h, protein kinases related to the GSK3β signaling pathway were analyzed by immunoblotting (Fig. 5A). Phosphorylation of PKA, ERK, p38, and SRC was not affected by melatonin treatment. However, melatonin treatment increased phosphorylation of AKT at serine 473 in a dose-dependent manner (Figs. 5A and 5B). Threonine 308 and serine 473 residue is present in the activation loop and C-terminal hydrophobic motif of AKT, respectively (*Manning & Toker, 2017*). Therefore,

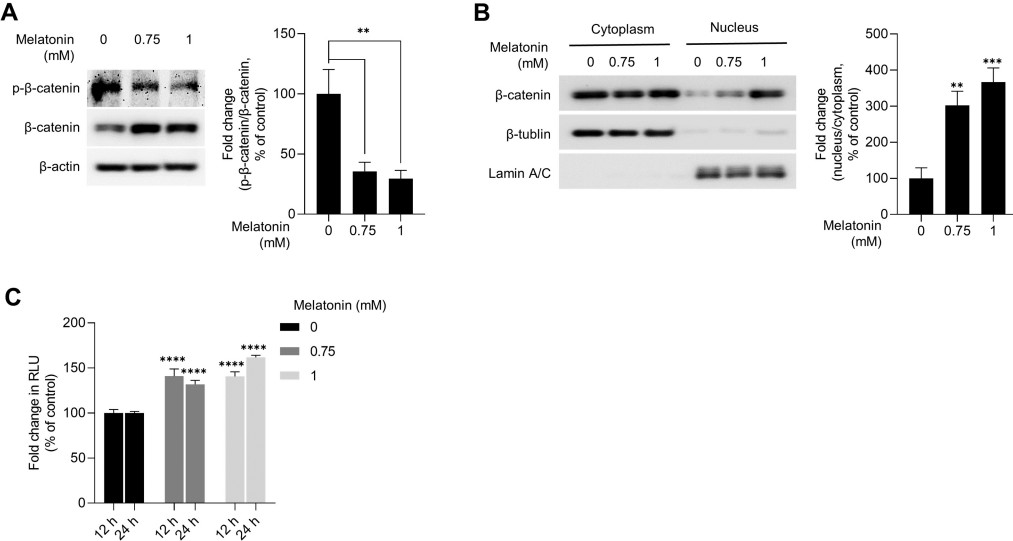

**Figure 4** **Effect of melatonin on β-catenin signaling pathway in HDP spheroids.** (A and B) HDP cells were treated with 0.75 and 1 mM melatonin for 24 h. (A) β-catenin stabilization was assessed by western blotting and β-actin served as a loading control. Quantification of the phosphorylation level was carried out using ImageJ software. (B) β-catenin translocation was assessed by western blotting. β-actin, Lamin C, and β-tubulin served as loading controls for total protein, nuclear fraction, and cytoplasmic fraction, re-spectively. Quantification of the phosphorylation level was carried out using ImageJ. (C) 293T cells were treated with 0.75 and 1 mM melatonin for 12 h and 24 h. TCF/LEF transcriptional activity was determined using a luciferase assay and normalizing luciferase activity to β-galactosidase activity. Data are presented as the mean ± SD of three independent experiments. (A and B) One-way ANOVA, followed by Turkey's post hoc test; ** $p < 0.01$, *** $p < 0.001$ *versus* DMSO treated control (A, $p = 0.0023$, and $p = 0.0014$; B, $p = 0.0012$, and $p = 0.0003$ respectively). (C) Two-way ANOVA, followed by Turkey's post hoc test; **** $p < 0.0001$ *versus* DMSO treated control (12 h, $p < 0.0001$, and $p < 0.0001$; and 24 h, $p < 0.0001$, and $p < 0.0001$ respectively).

we further analyzed the phosphorylation of threonine 308 residue and found that both threonine 308 and serine 473 residues on AKT were phosphorylated following melatonin treatment (Fig. S2).

To investigate whether melatonin-mediated phosphorylation of AKT is associated with activation of AKT, we used a lentivirus-mediated fluorescent translocation reporter system (HeLa/FoxO1-Clover) composed of fluorescent Clover protein fused with a modified FoxO1 protein, a well-characterized substrate of AKT (*Gross & Peter, 2015*). Cells were treated with melatonin (0, 200, 400 µM, and 2 mM) under serum starvation conditions for 24 h to inhibit AKT. Consequentially, melatonin treatment induced the subcellular localization of FoxO1-Clover from the nucleus to the cytoplasm (Fig. S3). These results suggest that melatonin enhanced the activation of AKT signaling, followed by the induction of the inhibitory phosphorylation of GSK3β at the serine 9 residue.

We further investigated whether AKT activation by melatonin induces phosphorylation of GSK3β at the serine 9 residue and regulates the expression of β-catenin using the selective PI3K/AKT inhibitor LY294002. Fig. 5C shows that treatment with LY294002 restored phosphorylation of AKT at serine 473, phosphorylation of GSK3β at serine 9,

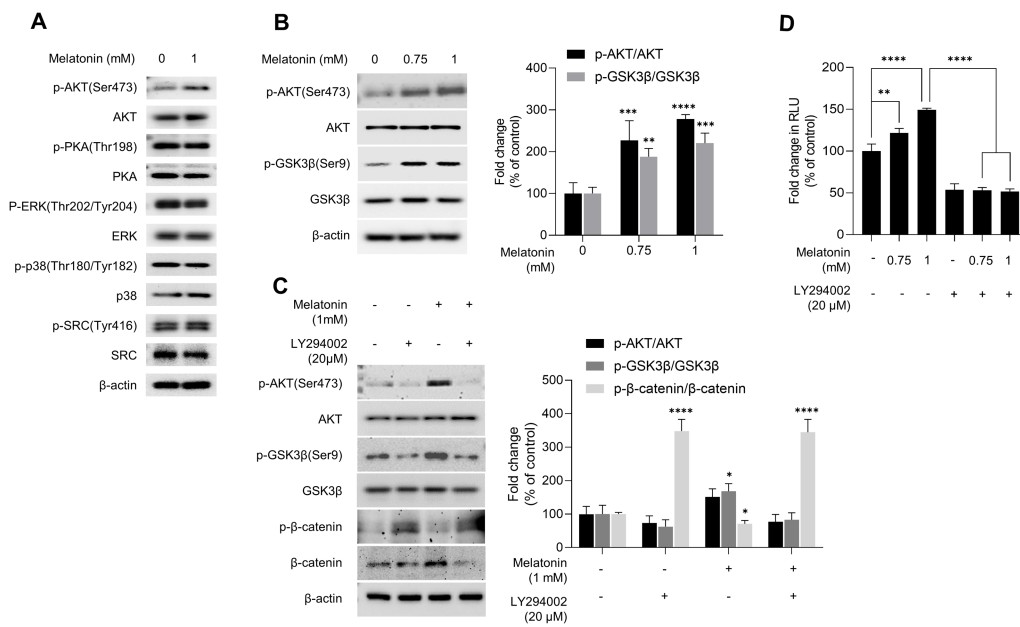

**Figure 5** **Effect of melatonin on AKT/GSK3β/β-catenin signaling in HDP spheroids.** (A) HDP cells were treated with 1 mM melatonin for 24 h. Protein levels of GSK3 β upstream signaling target genes were assessed by western blotting and β-actin served as a loading control. (B) HDP cells were treated with 0.75 and 1 mM melatonin for 24 h. The level of AKT/GSK3 β signaling phosphorylation were assessed by western blotting. β-actin served as a loading control. (C) HDP cells were treated with 1 mM melatonin with or without 20 μM LY294002 for 24 h. Phosphorylation of AKT/GSK3β/β-catenin signaling were assessed by western blotting and β-actin served as a loading control. Quantification of phosphorylated level was carried out using the ImageJ software. (D) 293T cells were treated with 1 mM melatonin with or without 20 μM LY294002 for 24 h. TCF/LEF transcriptional activity was determined using a luciferase assay by normalizing luciferase activity to β-galactosidase activity. Data are presented as the mean ± SD of three independent experiments. (B and C) Two-way ANOVA, followed by Turkey's post hoc test; $* p < 0.05$, $** p < 0.01$, $*** p < 0.001$, $**** p < 0.0001$ *versus* DMSO treated control (B: p-AKT/AKT, $p = 0.0002$, and $p < 0.0001$, and p-GSK3 β/GSK3β, $p = 0.0041$, and $p = 0.0003$; and C: p-GSK3 β/GSK3β; $p = 0.0103$, and p- β-catenin/β-catenin, $p < 0.0001$, and $p < 0.0001$ respectively). (D) One-way ANOVA, followed by Turkey's post hoc test; $** p < 0.01$, $**** p < 0.0001$ *versus* DMSO treated control ($p = 0.004$, and $p < 0.0001$ respectively) and, *versus* LY294002 untreated control ($p < 0.0001$, and $p < 0.0001$ respectively).

and increased β-catenin levels in sphere-cultured HDP cells. Furthermore, we confirmed that LY294002 treatment abolished melatonin-induced TCF/LEF-driven luciferase activity (Fig. 5D). Collectively, these results suggest that melatonin-induced inhibition of GSK3β and increase in β-catenin levels are mediated through the activation of the AKT signaling pathway in sphere-cultured HDP cells (Fig. 6).

## DISCUSSION

Melatonin acts as a ligand for melatonin receptors to mediate signal transduction (*Venegas et al., 2013*). Epidermal keratinocytes and dermal fibroblasts have been reported to express MTNR1A, whereas HF papillary fibroblasts express both MTNR1A and MTNR1B (*Slominski et al., 2003*; *Slominski, Wortsman & Tobin, 2005*). As expected, our results showed that the gene and protein expression levels of MTNR1A and MTNR1B in

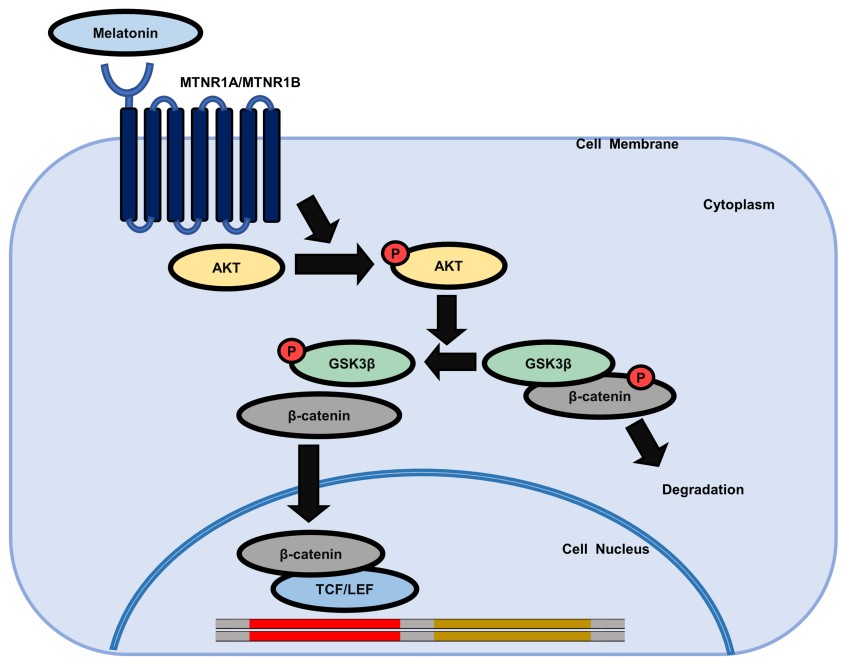

**Figure 6** **Schematic model of the possible mechanisms of the effect of melatonin in HDP cells.** Melatonin increases hair growth in human DP spheroids by activating the AKT/GSK3β/β-catenin signaling pathway *via* melatonin receptors.

HDP cells were higher than those in HEK and HDF cells. In particular, MTNR1B was expressed more significantly than MTNR1A in HDP. Previous reports have confirmed the proliferation of DP cells by investigating their size of 3D DP cells (*Choi et al., 2017*; *Lee et al., 2019*). We showed that melatonin increased the spheroid size of 3D HDP cells in a dose-dependent manner without inducing cytotoxicity. In addition, we found that 3D sphere culture conditions upregulated melatonin receptors, especially MTNR1B, suggesting that melatonin receptors can be restored in 3D HDP cells.

DP signature genes recovered by spheroid HDP cells are important regulators of hair inductivity (*Higgins et al., 2013*). Representative DP signature genes, including ALP, VCAN, BMP, and WNT, have been investigated as hair growth indicators. ALP is a highly conserved allosteric enzyme capable of hydrolyzing and phosphorylating various compounds. Although the precise physiological role of ALP in DP cells remains unknown, ALP activity is a marker for detecting the presence of DP cells and an indicator of hair inductivity (*Iida, Ihara & Matsuzaki, 2007*; *McElwee et al., 2003*; *Ohyama et al., 2010*; *Yang & Cotsarelis, 2010*). VCAN is specifically expressed in DP cells during the anagen phase and plays a crucial role in HF development (*Soma, Tajima & Kishimoto, 2005*; *Yang & Cotsarelis, 2010*). BMPs are members of the transforming growth factor-β superfamily and are abundantly present in hair bulbs and DP cells. BMP2 is a member of the BMP family that is vital for DP cell function (*Rendl, Polak & Fuchs, 2008*; *Yang & Cotsarelis, 2010*). WNTs are required to maintain epithelial stem cells and cell progeny during HF formation (*Rendl, Polak & Fuchs, 2008*). In particular, WNT5A, a well-known DP signature

gene in the hair bulb (*Rendl, Polak & Fuchs, 2008*), also mediates the second dermal signal for HF proliferation downstream of Sonic hedgehog (*Reddy et al., 2001*). We confirmed that DP signature genes were upregulated by melatonin treatment in a dose-dependent manner. Moreover, the DP signature genes that were upregulated by melatonin treatment were reduced by cotreatment with luzindole, a selective melatonin receptor antagonist (*Dubocovich, 1988*), with a greater affinity for MTNR1B than MTNR1A (*Browning et al., 2000*; *Dubocovich et al., 1998*). These results suggest that melatonin promotes hair growth by increasing the expression of DP signature genes, especially or at least in part through MTNR1B among other melatonin receptors.

The WNT/β-catenin signaling pathway is associated with hair-inductive properties in DP cells (*Kishimoto, Burgeson & Morgan, 2000*) and is an essential regulator of the expression of DP signature genes (*Iida, Ihara & Matsuzaki, 2007*; *McElwee et al., 2003*; *Ohyama et al., 2010*; *Reddy et al., 2001*; *Rendl, Polak & Fuchs, 2008*; *Soma, Tajima & Kishimoto, 2005*; *Yang & Cotsarelis, 2010*). In a murine model with loss-of-function of β-catenin, generation of HF was blocked and hair was completely lost after the first hair cycle, indicating that β-catenin plays a pivotal role in hair regrowth (*Enshell-Seijffers et al., 2010*). Various studies have demonstrated that the stabilization of β-catenin is involved in WNT-mediated hair regrowth as stabilized β-catenin interacts with TCF/LEF transcription factors and promotes the transactivation of hair growth-promoting genes (*Akiyama, 2000*; *DasGupta & Fuchs, 1999*). Moreover, melatonin treatment increased β-catenin expression in the hair bulbs in mice (*Uslu et al., 2014*) and promoted WNT10b and β-catenin gene expression in the skin of Inner Mongolia cashmere goats (*Liu et al., 2021*), suggesting that melatonin may affect β-catenin expression in skin cells, including DP cells. In the present study, as expected, we observed a decrease in β-catenin phosphorylation and an increase in total β-catenin levels and nuclear translocation of β-catenin following melatonin treatment in 3D HDP cells. Furthermore, melatonin was found to upregulate TCF/LEF-driven luciferase activity in a dose- and time-dependent manner. These results suggest that melatonin stabilizes β-catenin and activates the β-catenin signaling pathway in DP cells.

GSK3β is a major regulator of the canonical WNT signaling pathway as it determines the fate of β-catenin (*Kishimoto, Burgeson & Morgan, 2000*; *Yang & Cotsarelis, 2010*). GSK3β activity is regulated by its phosphorylation, which is mediated by various kinases, including AKT, PKA, ERK, p38, and SRC (*McCubrey et al., 2017*). Our results showed that melatonin enhanced the activation of AKT signaling, which induced the inhibitory phosphorylation of GSK3β at the serine 9 residue, whereas phosphorylation of PKA, ERK, p38, and SRC was not affected by melatonin treatment. Melatonin has been shown to induce AKT phosphorylation *via* melatonin receptors in primary astrocytes (*Kong et al., 2008*) and mouse N2A cells (*Beker et al., 2019*). We confirmed AKT activation by melatonin treatment in HeLa/FoxO-Clover cells, in which AKT activity can be visualized by FoxO1-tagged Clover fluorescent reporter. This result is consistent with the results of our experiment in DP cells and previous reports, thus supporting the hypothesis that melatonin induces AKT phosphorylation in 3D HDP cells. To further confirm this hypothesis, we conducted an immunoblot assay using HDP cells treated with melatonin and the PI3K inhibitor LY294002. We observed that melatonin increased the phosphorylation of AKT

at serine 473 residue and of GSK3β at serine 9 residue. Notably, increased β-catenin levels in 3D HDP cells were abolished by cotreatment with LY294002. Furthermore, LY294002 treatment reduced melatonin-induced TCF/LEF-luciferase activity. Taken together, these results suggest that melatonin-induced inhibition of GSK3β inhibition and increase in β-catenin levels are mediated through the activation of AKT signaling pathway in 3D HDP cells. Nuclear localization of β-catenin is regulated by various factors, including GSK3 β, APC, and p120 (*Gao, Xiao & Hu, 2014*; *Gu et al., 2016*). Although APC and p120 are β-catenin kinases, they regulate the nuclear localization of β-catenin. Therefore, further studies are required to elucidate the direct and/or indirect linkage between β-catenin stabilization and the expression of these factors.

Although our observations revealed the effects of melatonin in HDP cells, it is necessary to consider its effects on the entire HF system. In hair growth cycling, the crosstalk between mesenchymal and epithelial cells is complicated and sophisticated (*Sennett & Rendl, 2012*). Activation of the WNT/β-catenin signaling pathway in DP cells affects keratinocyte proliferation and differentiation (*Enshell-Seijffers et al., 2010*; *Kishimoto, Burgeson & Morgan, 2000*; *Soma et al., 2012*). Therefore, further studies on the interaction between DP cells and keratinocytes are needed to elucidate the hair growth inductive properties of melatonin in HFs *in vivo*.

## CONCLUSIONS

Although melatonin has been studied for many years for its effect on hair growth in humans and various mammals, the potential effects of melatonin on DP spheroids and its intracellular signaling mechanisms have not yet been investigated. We demonstrated that melatonin increased the volume of 3D spheroids of human DP cells and induced hair growth by activating the AKT/GSKSβ/β-catenin signaling pathway. Our results revealed that melatonin functions in human DP cells, which play an important role in hair growth in HFs.

### Funding
This paper was funded by Konkuk University in 2018. The funders had no role in study design, data collection and analysis, decision to publish, or preparation of the manuscript.

### Grant Disclosures
The following grant information was disclosed by the authors:
Konkuk University.

### Competing Interests
The authors declare there are no competing interests.

## Author Contributions

- Sowon Bae conceived and designed the experiments, performed the experiments, analyzed the data, prepared figures and/or tables, authored or reviewed drafts of the paper, and approved the final draft.
- Yoo Gyeong Yoon conceived and designed the experiments, performed the experiments, analyzed the data, authored or reviewed drafts of the paper, and approved the final draft.
- Ji Yea Kim conceived and designed the experiments, authored or reviewed drafts of the paper, and approved the final draft.
- In-Chul Park conceived and designed the experiments, authored or reviewed drafts of the paper, and approved the final draft.
- Sungkwan An conceived and designed the experiments, authored or reviewed drafts of the paper, and approved the final draft.
- Jae Ho Lee conceived and designed the experiments, authored or reviewed drafts of the paper, and approved the final draft.
- Seunghee Bae conceived and designed the experiments, authored or reviewed drafts of the paper, and approved the final draft.

## Data Availability

The raw figures and data are available in the Supplementary Files.

## Supplemental Information

Supplemental information for this article can be found online at http://dx.doi.org/10.7717/peerj.13461#supplemental-information.

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
