# Peer review of "Melatonin increases growth properties in human dermal papilla spheroids by activating AKT/GSK3β/β-Catenin signaling pathway"

_PeerJ, doi:10.7717/peerj.13461_

## Round 0.1 · original submission · Major Revisions

Dear Authors:

Please respond to the reviewers' comments point by point and then, resubmit it to the journal.

Reviewer 1 has suggested that you cite specific references. You are welcome to add it/them if you believe they are relevant. However, you are not required to include these citations, and if you do not include them, this will not influence my decision.

·

Basic reporting

The paper is of interest, however, it requires clarifications

METHODOLOGY
DMSO was used as a solvent, per figures legend. What was its concentration.
Please provide this information in the materials and methods.
MT2/1 antagonists, luzindoie was used. Please clarify, when it was added. Prior addition of melatonin? If yeas how many hours before. Note that you use 10-4-10-3 M range for melatonin and 50 uM for luzindole
Results
mM range of melatonin was used. How this relates to MT1 and MT2. For membrane bound receptors nM range would be expected. Is there an issue in the uptake of the ligand into spheroids?
Interpretation
You did not measure hair growth properties, you only measure behavior of DP spheroids. Please correct the title and interpretation
Discussion
Please discuss limitations which are obvious
Mention that skin cells produce melatonin (Slominski A, Pisarchik A, Semak I, Sweatman T, Wortsman J, Szczesniewski A, Slugocki G, McNulty J, Kauser S, Tobin DJ, Jing C, Johansson O (2002) Serotoninergic and melatoninergic systems are fully expressed in human skin. FASEB J 16, 896-898; full text FASEB J (April 23, 2002) 10.1096/fj.0.1-0952fje), which can undergo rapid metabolism) ((Kim TK, Kleszczynski, Janjetovic Z, Sweatman T, Lin Z, Li W, Reiter R, Fischer T, Slominski AT (2013) Metabolism of melatonin and biological activity of intermediates of melatoninergic pathway in human skin cells. FASEB J 27, 2742–2755: doi: 10.1096/fj.12-224691), issues highly relevant to the subject..
Mention the possibility of indirect action through its metabolite(s) (Exp Dermatol 26:563–568, doi: 10.1111/exd.13208. PubMed PMID: 27619234)
References

.

Experimental design

DMSO was used as a solvent, per figures legend. What was its concentration.
Please provide this information in the materials and methods.
MT2/1 antagonists, luzindoie was used. Please clarify, when it was added. Prior addition of melatonin? If yeas how many hours before. Note that you use 10-4-10-3 M range for melatonin and 50 uM for luzindole
Results
mM range of melatonin was used. How this relates to MT1 and MT2. For membrane bound receptors nM range would be expected. Is there an issue in the uptake of the ligand into spheroids?
Interpretation
You did not measure hair growth properties, you only measure behavior of DP spheroids. Please correct the title and interpretation

Validity of the findings

See basic reporting

Additional comments

See basic reporting

Reviewer 2 ·

Basic reporting

1 The English language should be improved to ensure that an international audience can clearly understand your text. Some examples where the language could be improved include lines 327, 424 – the current phrasing makes comprehension difficult. I suggest you have a colleague who is proficient in English and familiar with the subject matter review your manuscript, or contact a professional editing service.

Experimental design

Overall the experiments appear to be well performed

Validity of the findings

This area is of potential interest

Annotated reviews are not available for download in order to protect the identity of reviewers who chose to remain anonymous.

---

## Round 0.2 · accepted · Accept

The authors have fully responded to the reviewers' comments and the quality of the paper has significantly improved.

Reviewer 2 ·

Basic reporting

Now the revised manuscript has been adequately corrected and responded to my comments.

Experimental design

No comment

Validity of the findings

No comment